# Developmental and Nutritional Changes in Children with Severe Acute Malnutrition Provided with *n*-3 Fatty Acids Improved Ready-to-Use Therapeutic Food and Psychosocial Support: A Pilot Study in Tanzania

**DOI:** 10.3390/nu16050692

**Published:** 2024-02-28

**Authors:** Fredrick Cyprian Mwita, George PrayGod, Erica Sanga, Theresia Setebe, Gaudensia Joseph, Happyness Kunzi, Jayne Webster, Melissa Gladstone, Rebecca Searle, Maimuna Ahmed, Adolfine Hokororo, Suzanne Filteau, Henrik Friis, André Briend, Mette Frahm Olsen

**Affiliations:** 1Mwanza Research Centre, National Institute for Medical Research, Mwanza P.O. Box 1462, Tanzania; fredrick.mwita@nimr.or.tz (F.C.M.); erica.sanga@nimr.or.tz (E.S.); theresia.setebe@nimr.or.tz (T.S.); gaudensiajoseph61@gmail.com (G.J.); happykunzi@yahoo.com (H.K.); 2Faculty of Infectious and Tropical Diseases, London School of Hygiene and Tropical Medicine, Keppel Street, London WC1E 7HT, UK; jayne.webster@lshtm.ac.uk; 3Department of Women and Children’s Health, University of Liverpool, Alder Hey Children’s Hospital, Liverpool L12 2AP, UK; melglad@liverpool.ac.uk (M.G.); rebecca.searle@doctors.org.uk (R.S.); 4Department of Paediatrics, Bugando Medical Centre, Mwanza P.O. Box 1370, Tanzania; ahmedmaimuna@yahoo.co.uk (M.A.); adolfineh@gmail.com (A.H.); 5Faculty of Epidemiology and Population Health, London School of Hygiene and Tropical Medicine, Keppel Street, London WC1E 7HT, UK; suzanne.filteau@lshtm.ac.uk; 6Department of Nutrition, Exercise and Sports, University of Copenhagen, 2200 Copenhagen, Denmark; hfr@nexs.ku.dk (H.F.); andre.briend@gmail.com (A.B.); mette.frahm.olsen@regionh.dk (M.F.O.); 7Tampere Centre for Child, Adolescent and Maternal Health Research, Faculty of Medicine and Health Technology, Tampere University and Tampere University Hospital, Tampere University, Arvo Ylpön Katu 34, 33100 Tampere, Finland; 8Department of Infectious Diseases, Rigshospitalet, 2100 Copenhagen, Denmark

**Keywords:** severe acute malnutrition, child development, fatty acids, ready-to-use therapeutic foods, psychosocial stimulation

## Abstract

Children with severe acute malnutrition (SAM) are at high risk of impaired development. Contributing causes include the inadequate intake of specific nutrients such as polyunsaturated fatty acids (PUFAs) and a lack of adequate stimulation. We conducted a pilot study assessing developmental and nutritional changes in children with SAM provided with a modified ready-to-use therapeutic food and context-specific psychosocial intervention in Mwanza, Tanzania. We recruited 82 children with SAM (6–36 months) and 88 sex- and age-matched non-malnourished children. We measured child development, using the Malawi Development Assessment Tool (MDAT), measures of family and maternal care for children, and whole-blood PUFA levels. At baseline, the mean total MDAT z-score of children with SAM was lower than non-malnourished children; −2.37 (95% confidence interval: −2.92; −1.82), as were their total *n*-3 fatty acids, eicosapentaenoic acid (EPA) and docosahexaenoic acid (DHA) levels. After 8 weeks of intervention, MDAT z-scores improved in all domains, especially fine motor, among children with SAM. Total *n*-3 and EPA levels increased, total *n*-6 fatty acids decreased, and DHA remained unchanged. Family and maternal care also improved. The suggested benefits of the combined interventions on the developmental and nutritional status of children with SAM will be tested in a future trial.

## 1. Introduction

Severe acute malnutrition (SAM) affects almost 15 million children under the age of five in sub-Saharan Africa (SSA) [1]. According to a recent Demographic and Health Survey (DHS) 2023 [2], Tanzania reported that 3.3 per cent suffer from acute malnutrition (wasting or low weight-for-height), with the highest prevalence in lower socioeconomic groups. Malnutrition in children below five years of age is attributed to a combination of factors: maternal malnutrition, inadequate infant feeding practices, poor hygiene, and low-quality health care. Only 18.8% of breastfed children 6–23 months received a minimum acceptable diet, which has a major impact on growth and development. Malnutrition in children below five years of age in Tanzania continues to be a burden that requires a multisectoral approach and more context-specific interventions to avert long-term consequences. While ready-to-use therapeutic food (RUTF) improves the survival and nutritional recovery, cognitive development is known to often remain impaired after an episode of SAM [3,4].

During early life, brain development is influenced by both biological mechanisms and environmental factors such as nutritional intake, adequate stimulation, and responsive caregiving [5]. These factors can interplay with one another and all play a role in long-term child developmental outcomes [6].

One of the nutritional factors that have been considered important for children’s cognitive development is an adequate intake of polyunsaturated fatty acids (PUFAs) [7]. Previous studies have shown that children with SAM who were provided with RUTF continued to have suboptimal or even deteriorating PUFA levels after weight recovery [8,9]. Docosahexaenoic (C22:6*n*-3; DHA) and arachidonic acid (C20:4*n*-6; AA) are *n*-3 and *n*-6 long chain PUFAs, respectively, that are found in abundance in the cells of the central nervous system and play important roles in brain development [7]. AA and DHA can either be supplied by dietary sources, including breast milk and seafood or synthesised from their essential precursors, linoleic acid (C18:2*n*-6; LA) and alpha-linolenic acid (C18:3*n*-3; ALA), respectively [10]. However, large dietary amounts of LA may impair the conversion of ALA into DHA [11]. Consequently, the 2022 Codex Alimentarius guideline for RUTF now recommends lowering the upper limits for LA compared to previous recommendations [12,13].

In addition to adequate nutrition, psychosocial (PS) support is important to support the recovery of development in children treated for SAM [14,15,16]. Evidence from the 1970s showed that intense PS programmes, based on weekly home visits, benefited the development of children treated for SAM [17]. Thus, the World Health Organization (WHO) include PS in their ten-step guideline for hospital management of SAM [18]. Studies evaluating the effects of PS interventions in children treated for SAM have demonstrated an improvement in gross motor development, clinic attendance, and nutritional recovery [19,20,21,22,23]. However, the PS interventions conducted in these studies are often resource-intensive cannot be feasibly implemented in the community or scaled up for healthcare systems [17]. In practice, support for stimulation and responsive caregiving is rarely offered during SAM management in hospitals, and it is still not part of the guidelines for management in the community, where most children are now treated [13].

This paper describes the results from a pilot study of two interventions provided to children during the treatment of SAM to help improve cognitive and nutritional outcomes: RUTF with a PUFA composition modified to comply with the 2022 Codex guidelines and a context-specific PS intervention. We report changes in anthropometry, child development, whole-blood PUFAs, caregivers’ stimulation and support including family care indicators, caregiver–child interaction, and maternal depression after 8 weeks of combined intervention.

## 2. Materials and Methods

Our study consisted of three phases: (1) development of the intervention in partnership with local clinicians, Non-Governmental Organizations (NGOs), and caregivers of children with SAM, (2) testing the feasibility for delivering the two interventions (high oleic-RUTF and context-specific psychosocial intervention together, and (3) process evaluation, consisting of qualitative interviews and focus group discussions with caregivers and professionals concerning the acceptability and effects of the two interventions. In this paper, we described the quantitative data from phase 2 to evaluate the changes in child development, anthropometry, PUFAs, and caregivers’ stimulation, and support (additional information on the flowchart of the study, Appendix A).

Study design and setting: The BrightSAM trial development study was conducted in Mwanza, Tanzania, from June 2020 to February 2022. Our sample size calculation based on a stimulation trial conducted in Bangladesh that found an effect size of 0.52 SD [21]. We initially planned to recruit 200 participants: 100 children with SAM and 100 participants without SAM to allow us to detect an effect size of 0.44 SD with 80% power and 5% significant level. The main aim was to inform a future randomised 2-by-2 factorial trial that will investigate the individual and combined effects of modified RUTF and/or a context-specific PS intervention in improving developmental and nutritional outcomes in children treated for SAM. During the trial development study, we developed and piloted the RUTF and PS interventions and assessed the acceptability and implementation feasibility of these two interventions to be integrated into the existing SAM management in Tanzania and similar settings. The qualitative formative research that informed the design of the content and delivery mode of the PS intervention and mixed-methods evaluation of implementation fidelity and feasibility for future trials will be reported separately.

We recruited children with SAM from Bugando Medical Centre (BMC), a tertiary referral hospital for all districts and regional hospitals within Lake Zone, and 3 district hospitals: Misungwi, Nyamagana, and Buzuruga. We trained a network of community health workers in close contact with these hospitals to use mid-upper-arm circumference (MUAC) tapes for screening SAM in low-income communities within their catchment areas in the Mwanza region. An MUAC of <115 mm was regarded as SAM and any identified child was referred to the nearest district hospital for further clinical assessment and recruitment into the study if eligible.

Children with SAM who were hospitalised were included when transitioning to RUTF from F-75, while children with SAM without oedema, or infections, who passed the appetite test were directly started on RUTF through the outpatient management clinics of SAM. The study inclusion criteria were as follows: residence within Mwanza, confirmed SAM diagnosis as either MUAC < 115 mm, or weight-for-height (WHZ)-score < −3 as per WHO growth standards [24] or bilateral pitting oedema, children aged 6–36 months, and age of a caregiver giving consent ≥18 years. Children with SAM were excluded if they were allergic to peanuts or other RUTF ingredients or had any overt disability limiting the feasibility of delivering interventions or conducting assessments. Non-malnourished reference children (MUAC > 125 mm and WHZ > −2) were frequency-matched by age (+/−2 months) and sex to children with SAM and recruited from the same neighbourhoods to provide the reference levels of whole-blood PUFAs, child development, and family environment indicators.

RUTF composition and administration: the RUTF used in this study was produced by Nutriset, France, in accordance with Codex 2022 [12]. The RUTF was formulated using high-oleic peanuts as a main ingredient and contained 3.7 g/100 g of LA and 1.02 g/100 g of ALA. The composition contained no preformed DHA (additional information about nutritional composition, Appendix A). Each sachet contained 90 g of RUTF paste. RUTF was provided during the rehabilitation phase of SAM management in amounts dependent on the child’s body weight: 2 sachets/day for a child weighing 5–6.9 kg, 3 sachets/day for a child weighing 7–9.9 kg, and 4 sachets/day for a child weighing 10–14.9 kg [25].

Psychosocial intervention: We developed a short, focused, and context-specific PS intervention by adapting the WHO/UNICEF (United Nations Children’s Fund) Care for Child development package [26] based on qualitative research involving key stakeholders involved in treating children with SAM (details will be published separately). The adaptation was based on input from professionals in organisations engaged in early-childhood development such as the Tanzania Home Economics Organization (TAHEA) located in Mwanza, healthcare professionals, and caregivers of children with SAM. We piloted the developed intervention with the mothers of children with SAM and set up weekly group meetings, of which each child attended 8 at the research clinic of the National Institute for Medical Research in Mwanza. During these group sessions, on different weeks, mothers were taught about nutrition, the preparation of a balanced diet using locally available low-cost foods, water sanitation and hygiene (WASH), responsive parenting, and child stimulation. They were shown how to provide caregiver-led play and communication to their children and how to make toys from locally available materials.

Sociodemographic and clinical data: Upon the enrolment of both SAM and non-malnourished children, a research clinician collected data on sociodemographic characteristics including age, sex, child’s birth weight, parental education, occupation, marital status, and information on household assets from the Demographic Health Survey Tanzania [2]. Information about household assets was utilised to compute the socioeconomic status (SES) quintiles using principal component analysis [27]. Caregivers were also asked to report risk factors for developmental delays, such as hearing and visual deficits, and HIV exposure. HIV and malnutrition negatively reinforce each other. Children with HIV are at greater risk of malnutrition due to the low dietary intake because of mouth ulcers or loss of appetite, malabsorption of macro and micro-nutrients, and recurrent infections which increase the metabolic rate. Malnutrition impacts HIV infection through its negative effects on the immune system and a consequent increased risk of opportunistic infections [28]. Due to these factors, children with HIV and concurrent malnutrition tend to be at great risk of developmental problems [29,30].

Early-life stress measures: We assessed early-life adversities using an adapted version of a tool created by researchers in India which has demonstrated a high correlation with child development [31,32]. This tool was modified to remove items not relevant in the Tanzanian setting. The 16 adversities that we examined in the tool fell into three categories: (1) child stressors; (2) maternal stressors; and (3) socioeconomic adversities. Child stressors included: premature birth; child hospitalisation; inadequate supervision; and the separation of mother and child for more than a week. Maternal stressors included: the death of one or more of the mother’s close family members since becoming pregnant; the mother being seriously injured or sick since becoming pregnant; the mother being single, widowed, divorced, or separated; the mother ill or seriously injured during pregnancy; mother screened positive for mild, moderate or severe depression on the Patient Health Questionnaire (PHQ-9) in the past two weeks; and problematic alcohol/drugs use and/or selling by a person staying in the household. Socio-economic adversities included: family being in the lowest SES quintile; the mother or father having a very low level of education, i.e., no education, or only primary schooling; the father or mother having an irregular occupation, i.e., unemployed, seasonably employed, or casual labourer; and family debt or the inability to buy food for the family during the past six months.

Anthropometry: Anthropometric assessment was conducted by a trained research assistant. Weight was measured to the nearest 0.1 kg using a digital scale (ADE model M112600), and height/length was measured to the nearest 0.1 cm using a stadiometer/wooden height measuring board [33]. The measuring boards and scales were checked and calibrated regularly. MUAC was measured to the nearest 0.1 cm at the midpoint between the olecranon and the acromion process of the left arm using a non-elastic measuring tape. Anthropometric measurements were taken in triplicate and the median was used in analyses. The STATA package “zscore06” using the WHO standards was used to calculate WHZ and height-for-age (HAZ)-scores [34]. Stunting was defined as HAZ < −2.

Assessment of child development: Child development was assessed using the Malawi Development Assessment tool (MDAT) that was created and validated for use in African settings [35]. The assessments at baseline and 8 weeks follow-up were performed in a quiet room by trained research assistants with the caregiver present during the complete assessment. MDAT examines gross motor, fine motor, and language skills through direct observation and socio-emotional skills through caregivers’ reports. The tool has demonstrated good construct validity and sensitivity in predicting moderate-to-severe developmental delay in children from birth to 6 years of age [35]. MDAT z-scores for this study were based on reference data from a Malawian non-malnourished population [35].

Caregivers’ stimulation and support assessment: The Observation of Mother–Child Interaction measure (OMCI) [36], the Family Care Indicators (FCI) [37], and the PHQ-9 [38] were used in the study, as these measures were seen as potential mediators of intervention effects on developmental outcomes in children. OMCI is a 19-item observational tool which measures caregivers’ interaction behaviours (12 items) and child interaction behaviours (7 items). A caregiver was asked to play and talk with their child as they would normally, using a picture book provided, for five minutes without interruption. An observer counted each behaviour and then coded it as either 0 = never; 1 = very rarely; 2 = rarely; 3 = once in a while; and 4 = many times. All maternal items were summed to obtain maternal scores and child items were summed to obtain child scores. To reflect a positive and responsive mother–child interaction, the 4 negative items were reverse-coded to give a total score that ranged from 0 to 76 when maternal and child scores were combined. Before actual data collection, research assistants were trained as directed by Rasheed et al. [36] and attained sufficient inter-rater reliability (Kappa > 0.8).

We used FCI to collect information on (a) the availability and number of reading materials; (b) the availability and variety of play materials; and (c) family interaction in the home [37]. A raw score for each component was obtained by adding up positive answers to obtain score ranges of 0–3 for the source of playing materials, 0–7 for the variety of playing materials, and 0–6 for the family [39].

Depression symptoms among caregivers were assessed using the PHQ-9 [38], which is a 9-item depression screening tool. Responses for all 9 items are based on the caregiver’s experience during the past two weeks. Each item is scored on a 4-point scale. An example question is “Over the last two weeks how often have you been bothered by little interest or pleasure in doing things?” A total PHQ-9 score is calculated by adding up all individual item scores. A total score of 1–4 indicates minimal/no depression, 5–9 mild depression, 10–14 moderate depression, 15–19 moderately severe depression, and 20–27 severe depression.

Blood sampling and analysis of PUFAs: we sampled whole blood for fatty acids profiling as whole-blood fatty acid profiles result from the balanced proportions of fatty acids pools in plasma and cells [40]. A 1.5 mL sample of venous blood was collected in sodium citrate tubes from children with SAM at baseline and after 8 weeks and from non-malnourished children just once. Blood was transported to the laboratory at 2–8 °C. One mL of the venous sample was used to saturate 1 cm^2^ of a chromatography paper strip treated with 50 μg 2,6-di-tert-butyl-4-methylphenol (butylated hydroxytoluene) and 1000 μg deferoxamine mesylate salt (both from Sigma-Aldrich, St. Louis, MI, USA) for the analysis of whole-blood fatty acid composition. Data are given as a per cent by weight of individual fatty acids relative to the total fatty acid concentration in each sample (FA%). We regarded a high triene-to-tetraene ratio (defined as mead acid (C20:3*n*-9): AA (arachidonic acid, C20:4*n*-6) as an indicator of a low overall PUFA status [41]. Two indicators of low *n*-3 PUFA status were used: a high *n*-6 DPA (docosapentaenoic acid, C22:5*n*-6): DHA (docosahexaenoic acid, C22:6*n*-3) ratio, and a high *n*-6: *n*-3 PUFA ratio [42,43].

Data management and statistical analysis: Data were captured electronically using CSPro and analysed in STATA 17 (StataCorp, College Station, TX, USA). Descriptive data are presented as the mean ± standard deviation (SD), or number (%) as appropriate. Changes in anthropometry and child development at 8 weeks were assessed for both children with SAM and non-malnourished children using linear regression adjusted for sex, age, and month of inclusion to account for possible seasonal effects. Difference in the change between groups was also assessed. Mean levels of PUFAs, family care indicators, mother–child interaction, and maternal depression were compared between groups at baseline and changes in children with SAM and their families at 8 weeks were assessed using similar linear regression models.

## 3. Results

Recruitment occurred from June 2020 to February 2021. Due to recruitment challenges during the COVID-19 pandemic, we did not reach the original recruitment target (200 participants) but managed to enrol 170 participants: 82 children with SAM and 88 non-malnourished children. Among 82 children with SAM enrolled: 28 children received inpatient management of SAM with F-75 before transitioning to RUTF and the remaining 54 children started on RUTF directly as cases of outpatient management of uncomplicated SAM. During follow-up, 5 (3.0%) children with SAM died, and 7 (8.5%) were lost to follow-up. There were no deaths among non-malnourished children and 10 (11.4%) were lost to follow-up.

### 3.1. Background Characteristics of Children with SAM and Non-Malnourished Reference Children

Table 1 shows the baseline characteristics of children in the two groups. Children with SAM were more likely to be HIV-infected or exposed, to be from the lowest SES quintile, and to have mothers who never attended school. Children with SAM were 2 months younger (95% CI: −4.0, 0.1) than the non-malnourished children. More than half of the caregivers of children with SAM indicated concern that their child had a developmental delay, while this was not a concern to any of the caregivers of non-malnourished children. Families of children with SAM experienced early-life stress more than non-malnourished children. There was little difference between all children who enrolled at baseline and those who remained until the end of 8 weeks of intervention (Table 1).

### 3.2. Anthropometry

At baseline, children with SAM had a lower length/height and higher prevalence of stunting than non-malnourished children (95.1% vs. 56.8%). Among children with SAM having 8 weeks of data, both HAZ and WHZ scores increased during the intervention and 41 (55%) of those children with SAM who had follow-up data attained nutritional recovery that was defined as no SAM/MAM diagnosis using all three criteria used by WHO (oedema, WHZ, or MUAC) at 8-week intervention. Non-malnourished children gained considerably in HAZ and had a decline in the mean WHZ-score over the 8 weeks (Table 2).

### 3.3. Child Developmental Outcomes

Internal consistency for MDAT using Cronbach’s alpha was high for all domains in our setting (standardised Cronbach’s alpha of ≥0.85). At baseline, the children with SAM had much lower MDAT z-scores in all domains compared to the non-malnourished reference children (total MDAT score −2.37 (−2.92; −1.82)). After 8 weeks, improvement in all domains of development was observed in children with SAM when compared to their baseline, with the largest increase in the fine motor z-score 0.75 (0.46; 1.04). The MDAT z-scores of non-malnourished children also increased at end-line assessment, but less than the changes observed among children with SAM (Table 2).

### 3.4. PUFAs

Children with SAM had lower levels of total *n*-3 PUFA, DHA, and EPA but similar levels of *n*-6 PUFAs compared to non-malnourished children at baseline. After 8 weeks of intervention, the total *n*-3 PUFAs and EPA increased, the total *n*-6 PUFAs decreased, while DHA remained at the initial level (Table 3).

### 3.5. Caregivers’ Stimulation and Support

At baseline, children with SAM were less likely to have books or other play materials in their home and the FCI scores indicated that they interacted less with family members than non-malnourished children. The overall score for the observed interaction between mothers and children was also lower in the SAM group when compared to non-malnourished children (Table 4 and Appendix A).

In the eight week, the number and variety of playing materials had increased in the families of children with SAM and the observed interaction between mothers and children indicated some improvement.

Mothers of children with SAM had higher depression scores than the mothers of non-malnourished children at baseline, but at follow-up, these had decreased notably.

**Table 2 nutrients-16-00692-t002:** Anthropometry and child development in children with and without severe acute malnutrition.

	Baseline		Baseline Difference		Change at 8 Weeks		Change Difference	
	SAM, n = 70	Non-SAM, n = 78	SAM vs. Non-SAM		SAM, n = 70	Non-SAM, n = 78	SAM vs. Non-SAM	
	Mean (SD)	Mean (SD)	Mean (95% CI)	*p*	Mean (95% CI)	Mean (95% CI)	Mean (95% CI)	*p*
**Anthropometry**								
Weight ^1^, kg	5.6 (1.3)	9.6 (1.7)	−3.5 (−4.0; −3.1)	<0.001	1.0 (0.8; 1.2)	0.4 (0.2; 0.5)	0.8 (0.4; 0.9)	<0.001
Length/height, cm	63.9 (6.9)	73.7 (6.8)	−9.0 (−10.7; −7.3)	<0.001	3.2 (2.5; 4.0)	4.4 (3.7; 5.5)	−1.1 (−2.3; −0.02)	<0.001
MUAC, cm	10.7 (1.0)	14.8 (1.2)	−3.9 (−4.3; −3.5)	<0.001	1.7 (1.5; 1.9)	0.3 (0.1; 0.6)	1.4 (1.0; 1.7)	<0.001
HC, cm	41.8 (3.1)	45.5 (2.0)	−3.1 (−3.9; −2.3)	<0.001	1.5 (1.0; 2.0)	0.1 (−0.4; 0.5)	1.3 (0.7; 1.8)	<0.001
WHZ-score ^1^	−1.90 (1.94)	0.58 (1.15)	−2.28 (−2.90; −1.67)	<0.001	0.62 (0.16; 1.08)	−0.73 (−1.12; −0.34)	1.35 (0.71; 2.00)	<0.001
HAZ-score	−5.15 (1.80)	−2.16 (1.80)	−3.66 (−4.23; −3.10)	<0.001	0.60 (0.31; 0.89)	0.82 (0.55; 1.09)	−0.22 (−0.65; −0.21)	0.003
**Child development**								
Fine motor skills	−1.04 (1.82)	0.68 (1.22)	−1.71 (−2.28; −1.13)	<0.001	0.75 (0.46; 1.04)	0.16 (−0.12; 0.44)	0.59 (0.18; 0.99)	0.005
Gross motor skills	−1.57 (1.30)	0.32 (1.00)	−2.04 (−2.45; −1.62)	<0.001	0.32 (0.11; 0.53)	0.10 (−0.10; 0.30)	0.22 (−0.07; 0.51)	0.14
Language skills	−0.70 (1.45)	0.65 (1.14)	−1.38 (−1.85; −0.91)	<0.001	0.65 (0.39; 0.90)	0.40 (0.16; 0.64)	0.25 (−0.10; 0.60)	0.16
Socio-emotional skills	−1.22 (1.32)	−0.04 (1.16)	−1.06 (−1.53; −0.59)	<0.001	0.34 (0.09; 0.59)	0.65 (0.41; 0.88)	−0.31 (−0.65; 0.04)	0.08
Total MDAT score	−1.78 (1.75)	0.48 (1.25)	−2.37 (−2.92; −1.82)	<0.001	0.72 (0.51; 0.92)	0.42 (0.23; 0.61)	0.29 (0.02; 0.57)	0.038

Estimates are based on linear regression adjusted for age, sex, and month of enrolment. Results are shown for children with SAM and non-malnourished children (MUAC > 125 mm and WHZ > −2) who attended both baseline and 8-week visits. Children’s development was assessed using MDAT and reported as z-scores. ^1^ among children without oedema (n = 69). CI—confidence interval. HAZ—height for age z-score. HC—head circumference. MDAT—Malawi Development Assessment Tool. MUAC—mid-upper arm circumference. SAM—severe acute malnutrition. SD—standard deviation. WHZ—weight for height z-score.

**Table 3 nutrients-16-00692-t003:** Whole-blood polyunsaturated fatty acids in children with and without severe acute malnutrition.

	Baseline		Baseline Difference		Change at 8 Weeks	
	SAM, n = 70	Non-SAM, n = 72	SAM vs. non-SAM		(SAM at 8 Weeks vs. SAM at Baseline, n = 69)	
	Mean (SD)	Mean (SD)	Mean (95% CI)	*p*	Mean (95% CI)	*p*
**Whole-blood polyunsaturated fatty acids (PUFA)**						
Total *n*-3 PUFA, FA%	5.26 (1.58)	6.64 (1.44)	−1.79 (−2.36; −1.22)	<0.001	0.39 (0.13; 0.66)	0.004
Total *n*-6 PUFA, FA%	27.96 (2.58)	27.93 (1.94)	−0.37 (−1.25; 0.50)	0.40	−1.30 (−2.01; −0.59)	0.001
DHA (C22:6*n*-3), FA%	3.65 (1.42)	5.00 (1.23)	−1.69 (−2.18; −1.90)	<0.001	0.03 (−0.20; 0.25)	0.81
AA (C20:4*n*-6), FA%	8.58 (1.62)	9.61 (1.16)	−0.81 (−1.34; −0.29)	<0.001	0.60 (0.27; 0.93)	0.001
EPA (C20:5*n*-3) FA%	0.35 (0.18)	0.44 (0.19)	−0.12 (−0.19; −0.05)	0.001	0.10 (0.06;0.14)	<0.001
Indicator of low PUFA status						
Mead acid (C20:3*n*-9): AA ratio	0.02 (0.02)	0.01 (0.005)	0.01 (0.005; 0.02)	<0.001	0.002 (−0.002; 0.005)	0.37
Indicators of low *n*-3 PUFA status						
*n*-6 DPA (C22:5*n*-6):DHA	0.17 (0.11)	0.12 (0.07)	0.07 (0.04; 0.10)	<0.001	−0.01 (−0.03; 0.007)	0.19
*n*-6 PUFA: *n*-3 PUFA	5.79 (1.84)	4.44 (1.20)	1.48 (0.88; 2.08)	<0.001	−0.77 (−1.14; −0.39)	<0.001

Mean differences with 95% CI are based on linear regression adjusted for age, sex, and month of enrolment. Children with SAM are included at the initiation of nutritional treatment. Non-SAM children are referenced as non-malnourished children (MUAC > 125 mm and WHZ > −2). Whole-blood PUFA data are given in weight per cent relative to the total fatty acid concentration (FA%). The mean fatty acid concentration was 138 (SD 33) μg/100 μl whole blood. AA—arachidonic acid. CI—confidence interval. DHA—docosahexaenoic acid. DPA—docosapentaenoic acid. EPA—eicosapentaenoic acid. PUFA—polyunsaturated fatty acids. SAM—severe acute malnutrition. SD—standard deviation.

**Table 4 nutrients-16-00692-t004:** Family care indicators, mother–child interaction, and maternal depression in children with and without severe acute malnutrition.

	Baseline		Baseline Difference		Change at 8 Weeks	
	SAM, n = 70	Non-SAM, n = 78	SAM vs. Non-SAM		SAM at 8 Weeks vs. SAM at Baseline n = 70	
	Mean (SD)	Mean (SD)	Mean (95% CI)	*p*	Mean (95% CI)	*p*
**Family care indicators**						
Sources of playing materials (0–3 scale)	1.3 (0.6)	1.5 (0.5)	−0.3 (−0.5; −0.1)	0.004	0.8 (0.6; 0.9)	<0.001
Variety of playing materials (0–7 scale)	0.5 (0.8)	1.1 (1.5)	−0.9 (−1.3; −0.4)	<0.001	0.5 (0.1; 0.9)	0.01
Family interaction (0–6 scale)	2.7 (1.2)	3.4 (1.2)	−0.4 (−0.8; 0.03)	0.07	1.2 (0.9; 1.5)	<0.001
**Mother–child interaction**						
Mother and child overall score (0–76 scale)	34.8 (13.0)	43.1 (8.5)	−6.2 (−10.3; −2.1)	0.003	3.1 (−0.4; 6.5)	0.08
Mother score (0–48 scale)	24.3 (7.5)	28.5 (4.9)	−3.7 (−5.7; −1.7)	<0.001	1.6 (−0.4; 3.5)	0.11
Child score (0–28 scale)	10.6 (6.2)	14.6 (4.4)	−3.8 (−5.6; −2.1)	<0.001	1.5 (−0.2; 3.2)	0.08
**Maternal depression scale (PHQ9)**						
Summary score (0–27 scale)	9.0 (7.1)	4.2 (7.0)	4.2 (1.5; 7.0)	0.002	−7.6 (−9.3; −5.9)	<0.001

Mean differences with 95% CI are based on linear regression adjusted for age, sex, and month of enrolment. CI—confidence interval. PHQ9—Patient Health Questionnaire-9. SAM—severe acute malnutrition. SD—standard deviation. Mother–child interaction; the child score was obtained from 7 child interaction items and the mother score from 12 mother interaction items.

## 4. Discussion

This trial development study enabled us to develop and pilot the processes for testing an RUTF that complies with 2022 Codex guidelines [12] alongside an integrated context-specific PS intervention for children with SAM. Children with SAM improved their development scores over the 8 weeks of intervention with larger changes in MDAT z-scores than their non-malnourished comparison group. It is likely that this was due both to the impact of the interventions and because the SAM group had more development to “catch up” on after an episode of severe malnutrition.

A similar benefit for child development seen in a clinical trial conducted in Malawi testing high-oleic RUTF with additional preformed DHA that compared the 6-month MDAT scores of children treated for SAM versus their baseline scores [44]. This indicates that the improvements in child development scores seen in our study may be caused by the modified RUTF essential fatty acid profile. However, much of the improvement in the development of these SAM children may just be due to a general improvement in well-being for those children who were provided with RUTF. When children feel better, they are also more likely to perform better and score higher on developmental tests [45]. The individual and combined effects of the two interventions therefore need to be tested in a randomised controlled trial to be able to conclude on individual and combined effects.

At baseline, the levels of *n*-3 PUFA, DHA, and EPA were lower in children with SAM in our study sample (Table 3). However, these levels seemed relatively good when compared to the baseline PUFA levels among children with SAM/MAM studied elsewhere [42,46]. Looking at the baseline levels of DHA for instance, the values were more than twice that observed in children with SAM/MAM studied in other settings [42,46]. Our study population lives close to Lake Victoria with easy access to lake foods including fish.

We found improved levels of total *n*-3 fatty acid post-intervention similar to a trial in Malawi [44] that tested three different formulations of RUTF: high oleic-peanut RUTF with preformed DHA (DHA-HO-RUTF), high oleic-peanut RUTF without added preformed DHA (HO-RUTF), and standard RUTF (S-RUTF) on child development. In that trial, DHA-HO-RUTF, which contained 1.5 g/100 g DHA, was the only formulation that improved the level of DHA post-management. In our study, which used HO-RUTF without added preformed DHA, we observed that the DHA levels of children remained unchanged after supplementation. Previous studies with earlier versions of RUTFs have demonstrated levels of long-chain total *n*-3 fatty acids, EPA, and DHA that declined post-intervention [8,9,47]. Although one study suggested that lowering the dietary levels of LA may improve the synthesis of DHA [48], our findings suggest that additional preformed DHA from 1.5 g/100 g in HO-RUTF that tested in Malawi may be needed to increase the DHA levels post SAM management; however, the DHA encapsulation technique is recommended to obscure the taste of fish oil and to limit oxidative degradation that limits shelf life of RUTF. Our findings are in line with other studies that have shown that consuming RUTF with preformed DHA may be the solution that increases the level of DHA post-management in children with SAM [44,48,49].

The potential effects of PS programmes which promote responsive caregiving practices for children with SAM, are well described [17,22,50]. In practice, however, PS is often not provided during inpatient SAM treatment or within the community-based management of SAM. We developed a programme for PS that aimed to be feasible and accessible to provide in-hospital and clinic settings for children with SAM. This program encouraged and supported responsive and nurturing caregiving with more caregiver-led interactive play and communication, and the creation of play materials from locally available materials. In contrast to earlier studies that focussed on intensive family-based psychosocial intervention provided by trained personnel [17,20], our PS programme was designed to be self-sustainable by using the existing outpatient management of SAM to provide PS alongside RUTF supplementation and thus be integrated with the existing healthcare services.

The children admitted with SAM in our study had less caregiver stimulation and support than non-malnourished children—likely due to socioeconomic situations, low educational levels, and stressors within these families. Financial constraints and a lack of education may limit opportunities for positive parent–child interaction and may prevent children from having conducive environments to play. Many mothers of SAM children had increased signs of depression, which may be due to socio-economic constraints and other family issues, preventing them from being able to support their children in the way they would like [51]. These issues must be considered when providing PS programmes for children with SAM, as maternal depression and family stress may limit recovery or lead to re-admission unless tackled through social welfare and other support programmes.

This pilot study has several strengths. It included a population of children with SAM during their critical period of brain development (6–36 months) who were provided with a locally created feasible intervention linked with local organisations. We also conducted a detailed evaluation of anthropometry, child development, and other factors (socioeconomic status including some measures of adversities, mother–child interaction, and family care indicators) which may influence our intervention. Moreover, we included other factors (socioeconomic status, mother–child interaction, and family care indicators) which may influence our intervention. Furthermore, since Tanzania household food security varies greatly across the agricultural cycles of the year [52] we controlled for the season during analyses by incorporating the month of enrolment in the regression models. Despite these strengths, this was a trial development study and it was limited by having a small sample size, lack of randomisation or blinding, and lack of follow-up blood samples from non-malnourished children. We do not regard the 170 participants recruited instead of the planned 200 as a major limitation, as the included sample still provided sufficient variation to answer our objectives. Lastly, community screening was based on MUAC only. If we had also used WHZ and systematically assessed for nutritional oedemas, the studied sample would have represented the SAM population more broadly.

## 5. Conclusions

The combined interventions of a PUFA-modified RUTF and a context-specific PS intervention benefitted child development, reduced maternal depression, and increased the total *n*-3 PUFA, while the DHA levels were maintained. High-oleic peanuts are not likely to make a difference alone in improving DHA concentrations post-management of SAM if DHA is not added directly to the RUTF.

## Figures and Tables

**Table 1 nutrients-16-00692-t001:** Baseline characteristics of children with and without severe acute malnutrition.

	All Children			Children with 8 Weeks of Data		
	SAM, n = 82	Non-SAM, n = 88	*p*-Value	SAM, n = 70	Non-SAM, n = 78	*p*-Value
**Socio-demographic characteristics**						
Female, n (%)	43 (52.4)	42 (47.7)	0.54	39 (55.7)	38 (48.7)	0.40
Age, months (±standard deviation)	15.5 (6.9)	17.5 (7.9)	0.08	15.1 (6.7)	17.4 (7.8)	0.06
Parents’ marital status, n (%)			0.53			0.48
Married or co-habiting	47 (57.3)	57 (65.5)		38 (54.3)	49 (63.6)	
Divorced, separated, or widowed	13 (15.9)	12 (13.8)		11 (15.7)	11 (14.3)	
Never married	22 (26.8)	18 (20.7)		21 (30.0)	17 (22.1)	
Maternal education, n (%)			0.01			0.02
Never went to school	16 (20.2)	6 (7.3)		12 (17.7)	5 (6.9)	
Primary school	50 (63.3)	49 (59.8)		46 (67.7)	44 (61.1)	
Secondary school or higher ^1^	13 (16.5)	27 (32.9)		10 (14.7)	23 (31.94)	
Maternal occupation n (%)			<0.001			0.001
Salaried employment	3 (3.9)	1 (1.2)		2 (3.0)	1 (1.4)	
Petty trader (self-employed)	18 (23.4)	42 (51.9)		16 (24.2)	39 (54.9)	
Farmer (self-employed)	16 (20.8)	2 (2.5)		12 (18.2)	2 (2.8)	
Housewife/unemployed/student	40 (52.0)	36 (44.4)		36 (54.6)	29 (40.9)	
Household SES index, quintile, n (%)			0.002			0.01
Lowest	26 (31.7)	8 (9.1)		21 (30.0)	7 (9.0)	
2nd	17 (20.7)	17 (19.3)		16 (22.9)	16 (20.5)	
3rd	16 (19.5)	18 (20.5)		12 (17.14)	16 (20.5)	
4th	12 (14.6)	23 (26.1)		12 (17.1)	22 (28.2)	
Highest	11 (13.4)	22 (25.0)		9 (12.9)	17 (21.8)	
**Risk factors for child development delay ^2^ n** (%)						
Visual deficits	7 (8.5)	3 (3.4)	0.16	7 (10.0)	3 (3.9)	0.14
Hearing deficits	8 (9.8)	4 (4.6)	0.19	8 (11.4)	4 (5.1)	0.16
Delayed development	42 (51.2)	0 (0.0)	<0.001	32 (45.7)	0 (0.0)	<0.001
**HIV exposure** ^3^						
Child HIV-positive, n (%^2^)	7 (13.5)	0 (0.0)	<0.001	6 (8.6)	0 (0.0)	<0.001
Mother HIV-positive, n (%^2^)	17 (22.7)	2 (2.4)	<0.001	15 (21.4)	2 (5.6)	0.002
**Early-life stressors** ^4^, mean (SD)						
Overall adversity score (0–16 scale)	6.1 (2.2)	4.2 (1.9)	<0.001	6.2 (2.2)	4.3 (1.9)	<0.001
Child stressors (0–4 scale)	1.4 (0.9)	1.1 (0.8)	0.03	1.4 (0.9)	1.1(0.8)	0.08
Maternal stressors (0–6 scale)	1.7 (1.3)	1.0 (1.1)	0.0001	1.8 (1.4)	1.0 (1.1)	0.0002
Socio-economic stressors (0–6 scale)	3.0 (1.2)	2.1 (1.1)	<0.0001	3.0 (1.1)	2.1 (1.1)	<0.0001

Non-malnourished children have MUAC > 125 and WHZ > −2. ^1^ Any or completed, ^2^ Reported by a caregiver. ^3^ % of participants with known HIV status number; 55 children with SAM and 25 non-malnourished children and mothers; 75 mothers of children with SAM and 83 mothers of non-malnourished children. *p*-value: for group difference, continuous variables (student *t*-test), and categorical variables (chi-squared test). ^4^ Assessed using an adapted version of ELSQUS—Early Life Stress Questionnaire. HIV—Human Immunodeficiency Virus. MUAC—mid-upper arm circumference. n—number. SAM—severe acute malnutrition. SD—standard. SES—socio-economic status.

## Data Availability

The datasets used and analysed in this study are available to anyone for further analysis with approval from the Medical Research Coordinating Committee of the National Institute for Medical Research (NIMR), Tanzania.

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
