# Peer review of "Developmental and Nutritional Changes in Children with Severe Acute Malnutrition Provided with n-3 Fatty Acids Improved Ready-to-Use Therapeutic Food and Psychosocial Support: A Pilot Study in Tanzania"

_nutrients, 2024, doi:10.3390/nu16050692_

Round 1

Reviewer 1 Report

Comments and Suggestions for Authors

I think that the study is interesting but has some severe methodological limitations (i will focus more on clinical consequences and advantages and try to better explain methodology and the flowchart of the study, moreover a state of art of the pediatric nutrition situation in the country is necessary )

i think it is quite impossible to evaluate brain developement consequences in less than 2 years 

HIV patients are more often afected by SAM(i think this should be taken into account and better described)

Comments on the Quality of English Language

Moderate editing of English language required

Reviewer 2 Report

Comments and Suggestions for Authors

The article of Mwita and colleagues reported findings from a pilot study consisting of two interventions lasting 8 weeks provided to children with SAM to help improve cognitive and nutritional outcomes. The first intervention evaluated a ready-to-use therapeutic food (RUTF) formula, which was enriched with polyunsaturated fatty acids (PUFA) to meet CODEX Alimentarius's new guidelines 2022. The second intervention focused on psychosocial factors. The study focused on changes in anthropometry, child development, whole blood PUFAs, caregivers’ stimulation and support including family care indicators, caregiver-child interaction, and maternal depression.

The article is very useful as it complements current literature on the application of RUTF with an improved formulation containing pre-formed DHA along with a psychosocial intervention. Some details on the intervention are needed though. The following are comments to strengthen the presented work.

What was the sample size intended? Based on what power calculation?

Include some details of the psychosocial intervention like the length of the sessions. It says weekly interventions, this means there were 8 sessions?  The reviewer understands that this is a feasibility trial, but some description of the intervention is useful for readers to understand what you did.

LN328-339. It seems that the RUTF trial with higher DHA and EPA resulted in increased EPA, but not DHA. The discussion reflects some aspects of these findings, but this sentence requires finesse, “Our findings are in line with other studies that have shown that consuming RUTF with preformed DHA is the only way so far that increases the level of DHA post-management in children with SAM.” This reviewer wonders if effectively this is the only way. Maybe the only way so far.

Also, for discussion of PUFA outcomes, you might want to refer to the actual amount of DHA provided from your study vs. the others (e.g., Malawi) in terms of mg/sachet or 100g. This is because you indicated that an additional amount of DHA might be needed. Thus, what is this additional amount? From what initial amount? The combination of n-3 PUFAS is under 1 g per sachet. This makes this potential change quite feasible in terms of mass. However, more oxidizable oil might unduly affect stability during storage.  

You might want to include the difference in sampling whole blood vs. red blood cells for fatty acid profiling. For example, why whole blood and not RBCs.  

Round 2

Reviewer 2 Report

Comments and Suggestions for Authors

Dear Authors,

Thanks for considering all my reviews. You have addressed all comments.

Sincerely,

Reviewer 2